# MedficientSAM: A Robust Medical Segmentation Model with Optimized Inference Pipeline for Limited Clinical Settings

Bao-Hiep Le[1,2]*, Dang-Khoa Nguyen-Vu[1,2]*,
Trong-Hieu Nguyen-Mau[1,2], Hai-Dang Nguyen[1,2], and
Minh-Triet Tran[1,2,3]

[1] University of Science, Ho Chi Minh City, Vietnam
[2] Vietnam National University, Ho Chi Minh City, Vietnam
[3] John von Neumann Institute, Ho Chi Minh City, Vietnam
lbhiep20@apcs.fitus.edu.vn, nvdkhoa20@apcs.fitus.edu.vn,
20120081@student.hcmus.edu.vn, nhdang@selab.hcmus.edu.vn,
tmtriet@fit.hcmus.edu.vn

**Abstract.** Medical image segmentation plays a crucial role in clinical practice, aiding in identifying tumors, delineating organs, and monitoring disease progression. The advent of the Segment Anything Model (SAM) has enabled the development of universal medical image segmentation models that generalize across different modalities. However, the accessibility of such deep learning models in clinical settings is still limited by the reliance on powerful computing devices. In this paper, we propose MedficientSAM, which adopts the EfficientViT model to replace the heavy image encoder in SAM and then distills the knowledge from the MedSAM model on the challenge's training set. To further improve inference time, we re-implement the inference pipeline in the C++ programming language, optimizing the runtime on edge devices. MedficientSAM outperforms MedSAM in both accuracy and efficiency, achieving average DSC and NSD scores of **0.8642** and **0.8795**, respectively, on the public validation set. The average inference time is **1.0083 seconds** for 2D images and **8.9585 seconds** for 3D images. Our code and models are publicly available at https://github.com/hieplpvip/medficientsam.

**Keywords:** Medical image segmentation · Distillation · Embeddings Caching · C++ Implementation · Edge AI

## 1 Introduction

Medical image segmentation is a crucial clinical practice component, enabling precise diagnosis, treatment planning, and disease monitoring. Segmentation facilitates a deeper understanding of anatomical structures and abnormalities by delineating the boundaries of organs and pathological regions within images.

---

* The first two authors share equal contribution.

Early segmentation models for medical images were often based on the nnU-Net structure [7]. While effective, these models were limited to specific datasets, each tailored to a particular segmentation task. The emergence of the Segment Anything Model (SAM) [9], a generalized 2D segmentation model, has marked a significant paradigm shift in the segmentation task. A straightforward way to leverage the SAM model for medical images is to train it on a large-scale medical dataset [15]. Additionally, other approaches have been proposed, such as using adapters that allow the model to incorporate medical knowledge [22,5,3,10]. However, these efforts focus on adapting SAM to medical data while maintaining high computational demands. In most healthcare facilities, powerful computational devices are not available, and quick results are required, making it challenging to deploy these models in practice.

The "Segment Anything In Medical Images On Laptop" challenge aims to develop universal promptable medical image segmentation models that can be deployed on laptops or edge devices without relying on GPUs. Specifically, participants are tasked with creating a lightweight, bounding box-based segmentation model. The challenge also introduces a baseline model, LiteMedSAM, which replaces the heavy image encoder in MedSAM with TinyViT [23], a scaled-down vision transformer model using a progressive contraction approach [4]. The challenge provides a large training dataset with over one million image-mask pairs, covering 11 types of medical images, including Computed Tomography (CT), Magnetic Resonance Imaging (MRI), Positron Emission Tomography (PET), X-ray, ultrasound, mammography, Optical Coherence Tomography (OCT), endoscopy, fundus, dermoscopy, and microscopy, along with more than 20 types of cancer.

Many works have introduced lighter models to address computational constraints by replacing the heavy image encoder of SAM. In natural image processing, notable examples include MobileSAM [26] and EfficientViT-SAM [27]. MobileSAM utilizes TinyViT as a lightweight image encoder, similar to LiteMedSAM. EfficientViT-SAM, on the other hand, replaces traditional softmax attention [21] with lightweight ReLU linear attention [8], reducing computational complexity from quadratic to linear while maintaining functionality. The benchmarks in [27] indicate that EfficientViT-SAM offers higher throughput than MobileSAM, despite having more parameters, and also delivers superior segmentation accuracy, even outperforming the original SAM.

Given the advantages of EfficientViT-SAM over other lightweight models, we choose it as the student model and perform knowledge distillation from the MedSAM teacher model. The distillation focuses on the image encoder, employing the L2 loss function to align the outputs of the student and teacher encoders. The entire pipeline, called MedficientSAM, is then fine-tuned using a compound loss function, combining Focal loss [11] and Dice loss [18] in a 20:1 ratio. To further improve inference speed, we re-implement the inference process in C++. While Python is widely recognized for its ease of use and extensive libraries, it is relatively slow compared to lower-level programming languages. The optimized inference pipeline in C++ ensures that MedficientSAM can deliver high-speed

and accurate segmentation results even on resource-constrained devices, making it highly applicable in real-time clinical settings.

Our main contributions can be summarized as follows:

– We introduce MedficientSAM, utilizing knowledge distillation from the state-of-the-art segmentation model in medical images, MedSAM, and fine-tune it on a large-scale medical dataset to further improve accuracy.
– We implement a C++ inference pipeline to significantly reduce execution time on edge devices, offering a substantial performance boost over traditional Python-based pipelines.
– We propose the caching mechanism to reduce unnecessary recomputation of embeddings.

## 2  Method

### 2.1  Preprocessing

We follow the preprocessing in MedSAM implementation. For input images, we resize their longest dimension to align with the input size of EfficientViT-SAM's image encoder using bilinear interpolation, apply min-max scaling, and pad the resized images with zero values to create square dimensions (e.g., 512x512). The preprocessing steps for ground-truth masks are similar, except for the interpolation method and scaling approach. Specifically, we resize their longest dimension to match the input size of the image encoder using nearest-exact interpolation, then padding the resized masks with zeros to achieve square dimensions.

Additionally, since SAM only works on 2D images, for 3D volumes, we opt to slice them along the third dimension, creating 2D slices. These slices are then processed as described above.

### 2.2  Proposed Method

MedficientSAM is based on EfficientViT-SAM. We replace the image encoder from MedSAM with EfficientViT, a family of vision transformer models, while retaining the prompt encoder and mask decoder. Like EfficientViT-SAM, MedficientSAM has three variants, L0, L1, and L2, listed in increasing order of model sizes. Section 3 presents the speed-accuracy trade-offs analysis. Figure 1 demonstrates the macro architecture of EfficientViT-SAM-L1, which we use for official submission in the challenge. The model is trained in two stages: distillation and fine-tuning.

**Distillation** To initialize the image encoder, we transfer the knowledge of Med-SAM's image encoder (ViT-B) into EfficientViT through distillation. The goal is to align EfficientViT's and MedSAM-ViT-B's image embeddings by minimizing an L2 loss function.

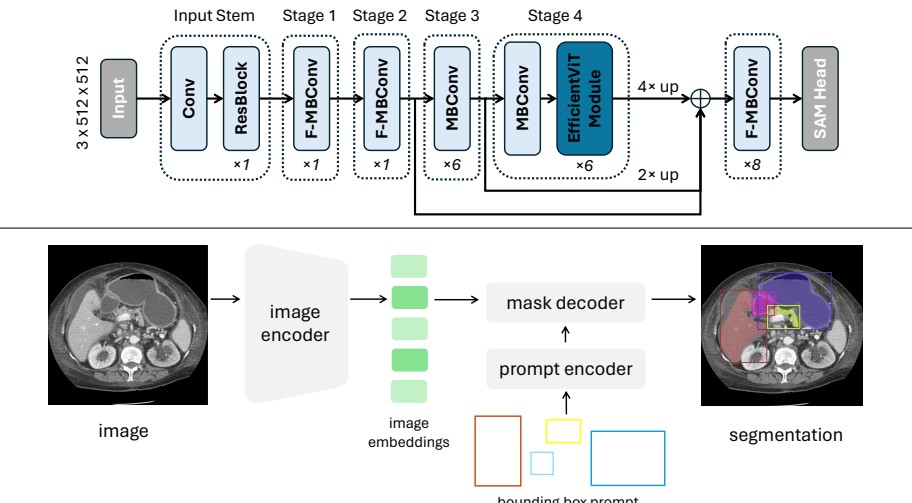

**Fig. 1. EfficientViT-SAM-L1's macro architecture (top) and MedficientSAM (bottom)**. *Top:* "ResBlock" refers to the basic building block from ResNet34 [6]. "F-MBConv" refers to the fused MBConv block from [20]. "EfficientViT Module" is the building block from [2]. *Bottom:* MedficientSAM is a promptable segmentation model that allows users to specify segmentation targets using bounding boxes.

**Fine-tuning** We integrate the distilled EfficientViT with MedSAM's pre-trained prompt encoder and mask decoder to create MedficientSAM. Subsequently, we perform end-to-end training to enhance performance further. To prompt the model, we generate box prompts by determining the smallest rectangles that cover the binary masks, introducing random shifts to improve the model's robustness.

**Loss function** Recently, compound loss functions have proven robust in various medical image segmentation tasks [14]. During fine-tuning, we use the weighted summation between Focal loss [11] and Dice loss [18] at a ratio of 20:1. Specifically, let $S$ and $G$ denote the result masks and ground truth, respectively. $N$ is the number of voxels in the image $I$. The focal loss is defined as

$$s_{t,i} = \begin{cases} s_i & \text{if } g_i = 1 \\ 1 - s_i & \text{otherwise} \end{cases} \tag{1}$$

$$L_{Focal} = -\frac{1}{N} \sum_{i=1}^{N} (1 - s_{t,i})^{\gamma} \log(s_{t,i}) \tag{2}$$

and dice loss is defined as

$$L_{Dice} = 1 - \frac{2\sum_{i=1}^{N} s_i g_i}{\sum_{i=1}^{N} s_i^2 + \sum_{i=1}^{N} g_i^2} \qquad (3)$$

The final loss $L$ is defined as

$$L = 20 \times L_{Focal} + L_{Dice} \qquad (4)$$

**3D Inference** Inspired by LiteMedSAM, for 3D volume inference, we start at the middle slice and propagate towards the ends, using the previously predicted mask slice as a guided prompt. We employ the idea of mask propagation as used in our previous work for organ segmentation [19]. If a binary mask is found in the previous slice, we obtain the bounding box that covers the binary mask and use it as the box prompt for the current slice instead of the box prompt from the input.

### 2.3   Post-processing

The binary masks output by MedficientSAM have a fixed size of $256 \times 256$. We first resize these output masks to match the input size of the image encoder, then crop out the padded zeros, and finally resize them back to their original resolution.

### 2.4   Inference optimization

While very convenient for model prototyping, Python is unsuitable for deployment due to its interpreting nature. We propose porting the pipeline to C++, a compiled language, and using OpenVINO as the model runtime to reduce inference time. Specifically, our inference optimization includes four parts:

- **Export model to OpenVINO format:** OpenVINO is an open-source deployment toolkit optimized to run on CPU. With OpenVINO's excellent support for PyTorch, we can easily export our model from PyTorch to OpenVINO format and run it in C++.
- **Port pre/post-processing stages to C++:** Unlike the model, the pre- and post-processing stages have to be ported to C++ manually. For image resizing, we use the OpenCV library [1]. For working with tensors, we use the xtensor library [16], which is inspired by NumPy.
- **Further optimization for compiled code:** To squeeze even more performance, we compile everything from source code to take advantage of optimizations like Advanced Vector Extensions and Link Time Optimization.
- **Embeddings caching for 3D volumes:** When inferring on 3D volumes with different prompting boxes, we need to iterate over the 2D slices and compute their embeddings repeatedly. Since the image encoder is the heaviest component of MedficientSAM, we propose caching the embeddings to avoid unnecessary recomputation.

## 3    Experiments

### 3.1    Dataset and evaluation measures

The dataset from the challenge is curated from publicly available sources, including some well-known datasets such as AbdomenCT-1K, AMOS, KiTS23, and COVID-19-20. The segmentation covers 11 medical image modalities (CT, MRI, PET, X-ray, ultrasound, mammography, OCT, endoscopy, fundus, dermoscopy, and microscopy) and targets more than 20 cancer types. The training set comprises over one million image-mask pairs, and the validation set includes about 30,000 image-prompt pairs.

The evaluation metrics include two accuracy measures: Dice Similarity Coefficient (DSC) and Normalized Surface Dice (NSD), alongside running time as an efficiency measure. These metrics collectively contribute to the ranking computation. The evaluation platform is CPU-only to simulate edge devices, running on an Intel(R) Xeon(R) W-2133 at 3.60GHz with 6 cores. Furthermore, the memory usage is constrained to a maximum of 8 GB. Participants are required to submit the solutions as Docker [17] containers.

### 3.2    Implementation details

**Environment settings:** Table 1 presents the development environments and requirements.

**Table 1.** Development environments and requirements.

| | |
|---|---|
| System | Ubuntu 22.04.3 LTS |
| CPU | AMD EPYC 7742 64-Core Processor |
| RAM | 256 GB |
| GPU | One NVIDIA A100 40G |
| CUDA version | 12.0 |
| Programming language | Python 3.10 |
| Deep learning framework | torch 2.2.2, torchvision 0.17.2 |

Docker containers are locally evaluated for their memory and time usage. The platform is detailed in Table 2. Constraints are set to simulate the official evaluation platform.

**Table 2.** Local evaluation platform.

| | |
|---|---|
| System | Ubuntu 22.04.3 LTS |
| CPU | Intel(R) Core(TM) i9-10900K |
| RAM | 8 GB |
| Docker version | 26.1.3 |

**Table 3.** Training protocols for distillation stage.

| | |
|---|---|
| Teacher Model | MedSAM-ViT-B[15] |
| Student Model | EfficientViT-L1[15] |
| Data augmentation | Horizontal Flipping and Vertical Flipping |
| Patch size | $512 \times 512 \times 3$ |
| Batch size | 8 |
| Total epochs | 8 |
| Optimizer | AdamW [13] with weight decay set to 0.0005 |
| Initial learning rate (lr) | 0.075 |
| Lr decay schedule | decay the Lr by 0.5 every epoch |
| Training time | 68 hours |
| Loss function | L2 |
| Number of model parameters | 43.59M |
| Number of flops | 49.23G |

**Training protocols:** We apply random horizontal and vertical flipping during the distillation stage for data augmentation. During fine-tuning, we apply Shift Scale Rotate in addition to flipping. We find that applying color-related augmentation techniques (such as RGB shift) reduces the accuracy. This is possibly due to medical image segmentation being sensitive to changes in color.

When distilling from MedSAM-Vit-B to EfficientViT, we need to repeatedly compute the output embeddings of MedSAM-Vit-B to use as labels for training EfficientViT. Since MedSAM-Vit-B is a very heavy model, this computation significantly contributes to the training time. One way to solve this is to precompute and save these embeddings to disk. However, due to the large size of the MedSAM dataset, precomputing the whole dataset would generate approximately 6 TB of embeddings, a very large amount of disk storage. Therefore, we resort to computing the embeddings on the fly and reducing the number of training samples to 400,000 randomly chosen image-mask pairs.

For the fine-tuning stage, the whole MedSAM dataset is used. Tables 3 and 4 detail the training protocols for the distillation and fine-tuning stages, respectively.

## 4    Results and discussion

### 4.1    Quantitative results

Table 5 compares the performance of the proposed model (MedficientSAM-L1) with the baseline model (LiteMedSAM) on the public validation set. We conduct ablation studies regarding the two-stage training process and the use of data augmentation.

Overall, LiteMedSAM scores highest on most targets, including CT, MR, US, and Fundus. In particular, MR and US showed significant gaps, with differences of 3% and 10%, respectively. However, LiteMedSAM falls far behind

**Table 4.** Training protocols for fine-tuning stage.

| Model | MedficientSAM-L1 |
|---|---|
| Data augmentation | Horizontal Flipping, Vertical Flipping, and Shift Scale Rotate |
| Patch size | $512 \times 512 \times 3$ |
| Batch size | 32 |
| Total epochs | 8 |
| Optimizer | AdamW [13] with default settings |
| Initial learning rate (lr) | 0.000002 |
| Lr decay schedule | Cosine Annealing [12] |
| Training time | 50.5 hours |
| Number of model parameters | 47.65M |
| Number of flops | 51.05G |

MedficientSAM and its variants for the remaining targets. Notably, with PET, LiteMedSAM achieved DSC and NSD scores of only 51.58% and 25.17%, respectively, while MedficientSAM achieved 73.00% and 58.03%, outperforming its ablated versions as well.

MedficientSAM has shown its effectiveness immediately after distillation, achieving DSC and NSD scores of 85.57% and 86.99% respectively, both higher than LiteMedSAM. Fine-tuning the whole pipeline improves the model's performance by 1-3% in several targets, except for ultrasound, which decreases by about 2% after fine-tuning. There is no significant difference in the version without augmentation; the two targets where it achieves the highest results, dermoscopy and microscopy, are only approximately equal to the randomly augmented MedficientSAM.

Generally, MedficientSAM achieves the highest average scores compared to the other methods, with DSC and NSD scores of 86.42% and 87.95%, respectively. The version without augmentation performs slightly better than the distillation-only version, and all three variants perform better than LiteMedSAM.

### 4.2   Qualitative results

Figure 2 illustrates several examples where the MedficientSAM performs well and some examples where it performs poorly. Specifically, in cases of good segmentation, selected examples include cell microscopy, chest x-ray, and abdominal endoscopy. The model performed very well in these examples, achieving DSC scores of 94%-98%, above the average. This may be because these images have high resolution, clear boundaries, and large object regions. Additionally, RGB images can be better segmented due to better color distinction than grayscale images.

In cases of challenging segmentation, the model performs much below average, at only 64%-68%. Selected examples include lesion PET scans, organ CT

**Table 5.** Quantitative evaluation results on the public validation set (top-1 scores are bolded). Ablation studies are performed to investigate the effectiveness of the fine-tuning stage and data augmentation.

| Target | LiteMedSAM | | Only Distillation | | No Augmentation | | MedficientSAM-L1 | |
|---|---|---|---|---|---|---|---|---|
| | DSC(%) | NSD(%) | DSC(%) | NSD(%) | DSC(%) | NSD (%) | DSC(%) | NSD (%) |
| CT | **92.26** | **94.90** | 91.13 | 93.75 | 92.24 | 94.71 | 92.15 | 94.80 |
| MR | **89.63** | **93.37** | 85.73 | 89.75 | 87.25 | 90.88 | 86.98 | 90.77 |
| PET | 51.58 | 25.17 | 70.49 | 54.52 | 72.05 | 56.26 | **73.00** | **58.03** |
| US | **94.77** | **96.81** | 84.43 | 89.29 | 81.99 | 86.74 | 82.50 | 87.24 |
| X-Ray | 75.83 | 80.39 | 78.92 | 84.64 | 79.88 | 85.73 | **80.47** | **86.23** |
| Dermoscopy | 92.47 | 93.85 | 92.84 | 94.16 | **94.24** | **95.62** | 94.16 | 95.54 |
| Endoscopy | 96.04 | 98.11 | **96.88** | **98.81** | 96.05 | 98.33 | 96.10 | 98.37 |
| Fundus | **94.81** | **96.41** | 94.10 | 95.83 | 94.16 | 95.89 | 94.32 | 96.05 |
| Microscopy | 61.63 | 65.38 | 75.63 | 82.15 | **78.76** | **85.22** | 78.09 | 84.47 |
| Average | 83.23 | 82.71 | 85.57 | 86.99 | 86.29 | 87.71 | **86.42** | **87.95** |

scans, and brain tumor MR scans. The poor performance of the model in these examples may be due to the characteristics of the image modality; for example, PET has a quite different color scale compared to other types. Additionally, in the second example, the segmented regions are separate from each other instead of being a single part, which can confuse the model. Furthermore, a low resolution can also make segmentation less effective, as the image will be blurry after resizing, and the boundaries will not be clear.

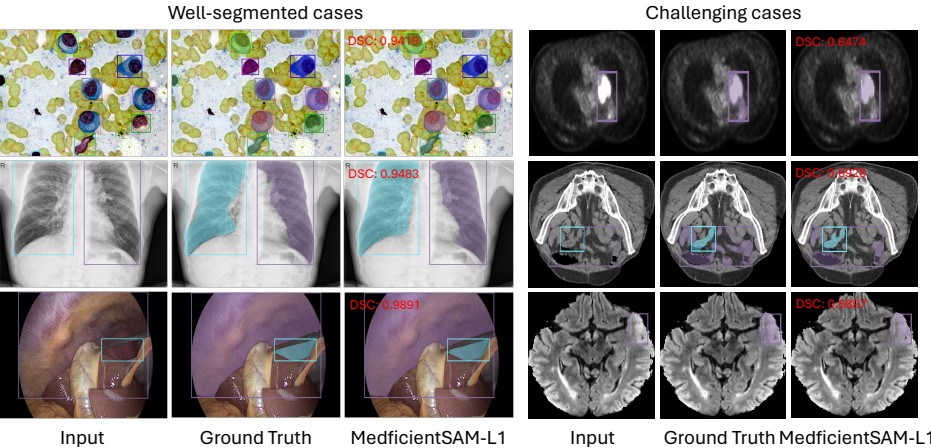

**Fig. 2.** Qualitative results from various public datasets. We illustrate both well-segmented and challenging examples for our proposed segmentation pipeline.

### 4.3   Segmentation efficiency results on validation set

Table 6 compares the efficiency of MedficientSAM against MedSAM and its smaller variant, LiteMedSAM, on the validation set. The average running time and memory usage are reported in seconds and megabytes. Although MedficientSAM has a higher resolution than LiteMedSAM (512 compared to 256), a higher number of FLOPs, and significantly more parameters, MedficientSAM still runs 5 times faster than LiteMedSAM. Regarding memory usage, MedficientSAM uses only half the amount of memory that LiteMedSAM requires. Moreover, MedficientSAM does not suffer from performance drops when switching to a lighter architecture, unlike LiteMedSAM. In fact, it even outperforms MedSAM, which is a heavier model with high resolution ($1024 \times 1024$). The superior performance of MedficientSAM demonstrates the effectiveness and robustness of our method.

**Table 6.** Segmentation efficiency results on the public validation set. The computational metrics are obtained on an Intel(R) Core(TM) i9-10900K, except for MedSAM, which can not run on CPU.

| Method | Res. | #Params | #FLOPs | Accuracy(%) | | Runtime | | Memory Usage | |
| --- | --- | --- | --- | --- | --- | --- | --- | --- | --- |
| | | | | DSC | NSD | 2D | 3D | 2D | 3D |
| MedSAM | 1024 | 93.74M | 488.24G | 84.91 | 86.46 | N/A | N/A | N/A | N/A |
| LiteMedSAM | 256 | 9.79M | 39.98G | 83.23 | 82.71 | 5.1 | 42.6 | 1135 | 1241 |
| MedficientSAM-L0 | 512 | 34.79M | 36.80G | 85.85 | 87.05 | **0.9** | **7.4** | **448** | **687** |
| MedficientSAM-L1 | 512 | 47.65M | 51.05G | **86.42** | **87.95** | 1.0 | 9.0 | 553 | 793 |
| MedficientSAM-L2 | 512 | 61.33M | 70.71G | 86.08 | 87.53 | 1.1 | 11.1 | 663 | 903 |

**Table 7.** Running time (s) of some validation cases, measured on an Intel(R) Core(TM) i9-10900K. "Baseline" refers to LiteMedSAM. "Proposed" refers to MedficientSAM-L1. "Ablation" refers to MedficientSAM-L1 running in Python. Embeddings caching is not used for "Baseline" and "Ablation".

| Case ID | Size | Num. Objects | Baseline | Ablation | Proposed |
| --- | --- | --- | --- | --- | --- |
| 3DBox_CT_0566 | (287, 512, 512) | 6 | 247.5 | 136.1 | 40.6 |
| 3DBox_CT_0888 | (237, 512, 512) | 6 | 70.1 | 37.8 | 15.0 |
| 3DBox_CT_0860 | (246, 512, 512) | 1 | 12.8 | 8.0 | 4.4 |
| 3DBox_MR_0621 | (115, 400, 400) | 6 | 107.0 | 56.8 | 14.9 |
| 3DBox_MR_0121 | (64, 290, 320) | 6 | 70.0 | 36.8 | 9.0 |
| 3DBox_MR_0179 | (84, 512, 512) | 1 | 12.5 | 7.3 | 3.9 |
| 3DBox_PET_0001 | (264, 200, 200) | 1 | 8.9 | 5.9 | 2.6 |
| 2DBox_US_0525 | (256, 256, 3) | 1 | 3.8 | 2.8 | 0.9 |
| 2DBox_X-Ray_0053 | (320, 640, 3) | 34 | 8.9 | 3.6 | 1.3 |
| 2DBox_Dermoscopy_0003 | (3024, 4032, 3) | 1 | 6.9 | 3.0 | 1.1 |
| 2DBox_Endoscopy_0086 | (480, 560, 3) | 1 | 4.3 | 2.8 | 1.0 |
| 2DBox_Fundus_0003 | (2048, 2048, 3) | 1 | 5.1 | 2.8 | 1.0 |
| 2DBox_Microscope_0008 | (1536, 2040, 3) | 19 | 14.2 | 3.4 | 1.3 |
| 2DBox_Microscope_0016 | (1920, 2560, 3) | 241 | 21.3 | 9.8 | 6.6 |

For more detailed analysis, Table 7 presents the running time of various cases from the public validation set. We conduct an ablation study on the speed-up of the C++ inference pipeline compared to Python. Note that LiteMedSAM, the baseline, is also running in Python. The ablation study also demonstrates the superiority over LiteMedSAM, with runtime reduced by more than half. This indicates that MedficientSAM's robustness is not only due to the optimized inference pipeline but also the architecture itself. Reimplementing the inference process in C++ further halves the running time.

### 4.4    Results on final testing set

| Team | Rank | Score | CT | MR | Endo | US |
|---|---|---|---|---|---|---|
| seno | 1 | 4.74 | 9/2/2 | 8/6/2 | 1/1/1 | 1/1/1 |
| automlfreiburg | 2 | 7.04 | 1/1/1 | 2/4/1 | 9/9/2 | 19/20/2 |
| skippinglegday | 3 | 7.11 | 2/3/6 | 7/12/6 | 12/12/7 | 2/2/7 |
| lkeb | 4 | 9.19 | 3/6/8 | 6/1/8 | 10/10/11 | 6/7/9 |
| yangspalworld | 5 | 9.22 | 8/9/3 | 16/18/3 | 15/15/4 | 9/9/3 |
| cvhci | 6 | 9.28 | 18/19/5 | 18/15/5 | 2/2/6 | 12/12/6 |
| organagent | 7 | 9.39 | 17/18/12 | 13/16/13 | 4/2/9 | 4/5/11 |
| hmi306 | 8 | 10.26 | 6/8/17 | 15/19/22 | 7/7/14 | 3/4/12 |
| hawken50 | 9 | 10.37 | 21/20/11 | 4/21/14 | 5/5/3 | 17/17/4 |
| uestcsd | 9 | 10.37 | 4/5/4 | 3/2/4 | 21/21/5 | 14/15/5 |

| Team | X-Ray | Fundus | Microscope | PET | OCT |
|---|---|---|---|---|---|
| seno | 8/8/2 | 1/1/1 | 7/8/1 | 18/16/3 | 9/8/2 |
| automlfreiburg | 15/15/1 | 15/15/2 | 11/11/2 | 5/7/2 | 7/7/4 |
| skippinglegday | 9/9/7 | 9/9/8 | 4/5/7 | 14/14/7 | 2/2/7 |
| lkeb | 19/19/10 | 12/12/12 | 10/10/9 | 4/4/11 | 11/11/9 |
| yangspalworld | 12/12/3 | 13/13/4 | 17/17/3 | 8/9/4 | 8/9/5 |
| cvhci | 1/1/6 | 4/4/6 | 13/14/6 | 12/15/1 | 23/23/1 |
| organagent | 6/5/9 | 3/3/9 | 1/1/11 | 23/22/14 | 5/6/11 |
| hmi306 | 5/6/11 | 5/5/14 | 3/3/15 | 22/23/19 | 1/1/10 |
| hawken50 | 10/11/4 | 8/8/3 | 2/2/5 | 3/3/23 | 18/17/3 |
| uestcsd | 21/21/5 | 20/20/5 | 19/19/4 | 2/1/5 | 14/14/6 |

**Table 8.** Ranking on the final testing set. Our team, "seno", achieved the top rank in the challenge. For each modality, the numbers denote the ranks of DSC, NSD, and Runtime, respectively.

On the final testing set, MedficientSAM ranks highest, outperforming all other teams in both segmentation accuracy and runtime. Our method achieved an average rank of 4.74, significantly ahead of the second-ranked team, which had an average rank of 7.04. Table 8 clearly demonstrates our dominance, particularly in the critical metric of runtime efficiency. Except for PET, we ranked

1st or 2nd in terms of running time across modalities, competing closely with the second-ranked team. However, unlike the second-ranked team, MedficientSAM maintains high accuracy across most targets (except for PET) on DSC and NSD metrics. While the second-ranked team falls to rank 20 in some targets, our method consistently performs well. MedficientSAM has set a new benchmark for speed and performance, significantly reducing computational time while maintaining high accuracy across various medical imaging modalities.

### 4.5   Limitation and future work

MedficientSAM has shown significant improvements but still has limitations and areas for future work. Currently, 3D volumes are processed independently on each slice, resulting in longer processing times and ignoring the spatial relationships between adjacent slices. This could potentially impact the accuracy of segmentations in 3D medical imaging. Besides, training with larger datasets, such as SA-Med2D-20M [25], could improve robustness and generalizability. Moreover, implementing model pruning and quantization could further reduce computational requirements. Finally, expanding MedficientSAM to segment all relevant structures within a medical image (i.e., Segment Everything) will enhance its applicability in diverse clinical scenarios.

## 5   Conclusion

In this work, we present MedficientSAM, leveraging EfficientViT to enhance both the efficiency and accuracy of MedSAM. Our method employs a two-stage training process, resulting in improved segmentation accuracy compared to MedSAM while substantially lowering computational demands. Additionally, we have developed a novel C++ inference pipeline, enabling MedficientSAM to operate on resource-constrained devices commonly found in clinical environments. We have open-sourced our code and models on GitHub to foster further research and collaboration.

**Acknowledgements** This research is supported by Vingroup Innovation Foundation (VINIF) in project code VINIF.2019.DA19. We thank University of Science, VNU-HCM, for providing access to the DGX A100 server used in this work. We thank all the data owners for making the medical images publicly available and CodaLab [24] for hosting the challenge platform.

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
