# OpenReview forum: "MedficientSAM: A Robust Medical Segmentation Model with Optimized Inference Pipeline for Limited Clinical Settings"
_thecvf.com/CVPR/2024/Workshop/MedSAMonLaptop — CVPR24 MedSAMonLaptop_

### Official Review · Reviewer_UxLo · 2024-06-14
**Reproducibility and compleness are good, with few unclear points and format problems; strongly recommend to accept.**

**Rating:** 8
**Confidence:** 4

**Review:**

The paper proposes an excellent end-to-end pipeline from training to deployment. The accuracy and efficiency of their method is great. The source code provides comprehensive reproducibility. Here are some advices for the paper:

1. In Fig.1, the author shows that the output of the image encoder is the image embeddings. Correspondingly, the output of the prompt encoder should be pointed out as well.

2. In Fig.2, the purple masks and boxes on the right subfigure is inapparent. The author should use a brighter color.

3. The font in figures should be Times New Roman.

4. In Table.6, the reason that MedSAM gets N/A on both runtime and memory usage should be pointed out. The reader may not be one of the challenge attendee and not familiar with the threshold.

5. In section 2.2 Loss function, the author should provide further explanation as to why the Focal Loss deserves more attention.

---

### Official Review · Reviewer_9MnF · 2024-06-16
**Good performance in segmentations, and significant improvement over speed with OpenVINO implementations**

**Rating:** 8
**Confidence:** 4

**Review:**

Image encoder is replaced with EfficientVit, and distilled from the original baseline. Validation results seem to rank among the top 10.

The C++ implement using OpenVino seems to significantly improves over the baseline.

One concern seems that the C++ implementation needs custom built on the authors' computer, whether that works the same in the docker in the test will affect the final results.

Some drawbacks:

In Fig 1 upper graph, MBC and F-MBC blocks are not explained in this paper.
In Fig 1 lower graph, the algorithm framework has no difference with the baseline.

In loss function, how did you derive the 20:1 ratio of Focal loss and dice loss?

---

### Official Review · Reviewer_WyBD · 2024-06-16
**Review for 'MedficientSAM: A Robust Medical Segmentation Model with Optimized Inference Pipeline for Limited Clinical Settings'**

**Rating:** 9
**Confidence:** 4

**Review:**

Pros:
- Well-written, clear description of the strategy and methodology used.
- Figures support the understanding of the approach clearly.
- Efficient pipeline compiled in C++.

Missing information:
- Distillation details regarding the prompt encoder and mask decoder are not given.

Problems:
- The text font in figures is not Times New Roman.
- Typo in section 2.4: Further instead of Futher.
- Missing vertical lines for tables in the **Experiments** section.

---

### Decision · Program_Chairs · 2024-10-01

Accept